# Approximate Inference Turns Deep Networks into Gaussian Processes

**Mohammad Emtiyaz Khan**
RIKEN Center for AI Project
Tokyo, Japan
`emtiyaz.khan@riken.jp`

**Alexander Immer**[*][†]
EPFL
Lausanne, Switzerland
`alexander.immer@epfl.ch`

**Ehsan Abedi**[*][†]
EPFL
Lausanne, Switzerland
`ehsan.abedi@epfl.ch`

**Maciej Korzepa**[*][†]
Technical University of Denmark
Kgs. Lyngby, Denmark
`mjko@dtu.dk`

## Abstract

Deep neural networks (DNN) and Gaussian processes (GP) are two powerful models with several theoretical connections relating them, but the relationship between their training methods is not well understood. In this paper, we show that certain Gaussian posterior approximations for Bayesian DNNs are equivalent to GP posteriors. This enables us to relate solutions and iterations of a deep-learning algorithm to GP inference. As a result, we can obtain a GP kernel and a nonlinear feature map while training a DNN. Surprisingly, the resulting kernel is the neural tangent kernel. We show kernels obtained on real datasets and demonstrate the use of the GP marginal likelihood to tune hyperparameters of DNNs. Our work aims to facilitate further research on combining DNNs and GPs in practical settings.

## 1 Introduction

Deep neural networks (DNN) and Gaussian processes (GP) models are both powerful models with complementary strengths and weaknesses. DNNs achieve state-of-the-art results on many real-world problems providing scalable end-to-end learning, but they can overfit on small datasets and be overconfident. In contrast, GPs are suitable for small datasets and compute confidence estimates, but they are not scalable and choosing a good kernel in practice is challenging [3]. Combining their strengths to solve real-world problems is an important problem.

Theoretically, the two models are closely related to each other. Previous work has shown that as the width of a DNN increases to infinity, the DNN converges to a GP [4, 5, 13, 16, 22]. This relationship is surprising and gives us hope that a practical combination could be possible. Unfortunately, it is not clear how one can use such connections in practice, e.g., to perform fast inference in GPs by using training methods of DNNs, or to reduce overfitting in DNNs by using GP inference. We argue that, to solve such practical problems, we need the relationship not only between the models but also between their training procedures. The purpose of this paper is to provide such a theoretical relationship.

We present theoretical results aimed at connecting the training methods of deep learning and GP models. We show that the Gaussian posterior approximations for Bayesian DNNs, such as those obtained by Laplace approximation and variational inference (VI), are equivalent to posterior distributions of GP regression models. This result enables us to relate the solutions and iterations of a deep-learning algorithm to GP inference. See Fig. 1 for our approach called DNN2GP. In addition,

---

[†]Equal contribution. [*]This work is performed during an internship at the RIKEN Center for AI project.

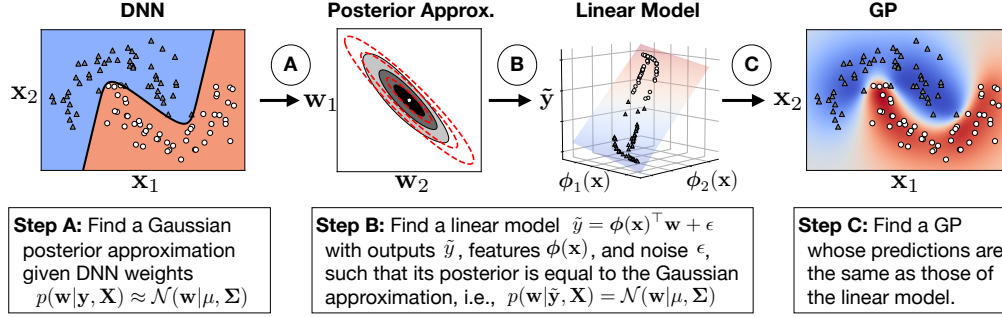

**Step A:** Find a Gaussian posterior approximation given DNN weights $p(\mathbf{w}|\mathbf{y}, \mathbf{X}) \approx \mathcal{N}(\mathbf{w}|\mu, \Sigma)$

**Step B:** Find a linear model $\tilde{y} = \phi(\mathbf{x})^\top \mathbf{w} + \epsilon$ with outputs $\tilde{y}$, features $\phi(\mathbf{x})$, and noise $\epsilon$, such that its posterior is equal to the Gaussian approximation, i.e., $p(\mathbf{w}|\tilde{\mathbf{y}}, \mathbf{X}) = \mathcal{N}(\mathbf{w}|\mu, \Sigma)$

**Step C:** Find a GP whose predictions are the same as those of the linear model.

Figure 1: A summary of our approach called DNN2GP in three steps.

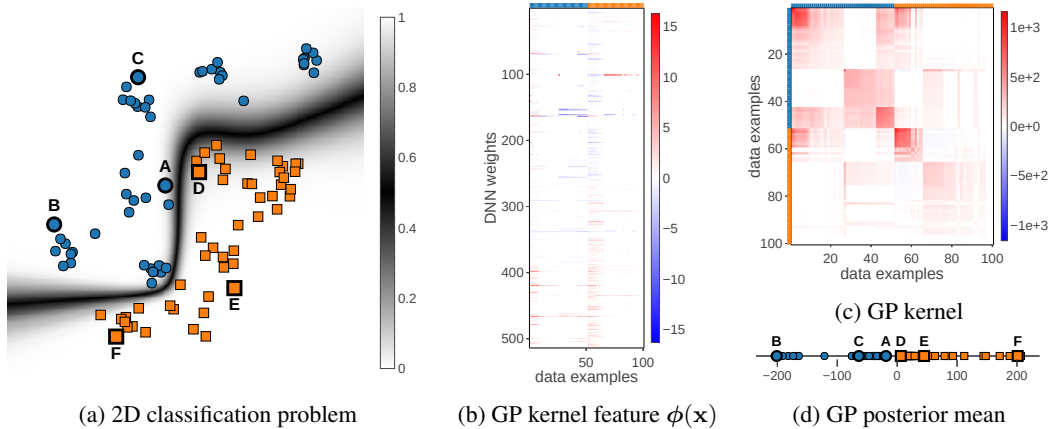

(a) 2D classification problem  (b) GP kernel feature $\phi(\mathbf{x})$  (c) GP kernel / (d) GP posterior mean

Figure 2: Fig. (a) shows a 2D binary-classification problem along with the predictive distribution of a DNN using 513 parameters. The corresponding feature and kernel matrices obtained using our approach are shown in (b) and (c), respectively (the two classes are grouped, and marked with blue and orange color along the axes). Fig. (d) shows the GP posterior mean where we see a clear separation between the two classes. Surprisingly, the border points A and D in (a) are also at the boundary in (d).

we can obtain GP kernels and nonlinear feature maps while training a DNN (see Fig. 2). Surprisingly, a GP kernel we derive is equivalent to the recently proposed neural tangent kernel (NTK) [8].We present empirical results where we visualize the feature-map obtained on benchmark datasets such as MNIST and CIFAR, and demonstrate their use for DNN hyperparameter tuning. The code to reproduce our results is available at `https://github.com/team-approx-bayes/dnn2gp`. The work presented in this paper aims to facilitate further research on combining the strengths of DNNs and GPs in practical settings.

## 1.1 Related Work

The equivalence between infinitely-wide neural networks and GPs was originally discussed by Neal [16]. Subsequently, many works derived explicit expressions for the GP kernel corresponding to neural networks [4, 7, 16] and their deep variants [5, 6, 13, 18]. These works use a prior distribution on weights and derive kernels by averaging over the prior. Our work differs from these works in the fact that we use the *posterior* approximations to relate DNNs to GPs. Unlike these previous results, our results hold for DNNs of finite width.

A GP kernel we derive is equivalent to the recently proposed Neural Tangent Kernel (NTK) [8], which is obtained by using the Jacobian of the DNN outputs. For randomly initialized trajectories, as the DNN width goes to infinity, the NTK converges in probability to a deterministic kernel and remains asymptotically constant when training with gradient descent. Jacot et al. [8] motivate the NTK by using kernel gradient descent. Surprisingly, the NTK appears in our work with an entirely different approach where we consider approximations of the posterior distribution over weights. Due

to connections to the NTK, we expect similar properties for our kernel. Our approach additionally shows that we can obtain other types of kernels by using different approximate inference methods.

In a recent work, Lee et al. [14] derive the mean and covariance function corresponding to the GP induced by the NTK. Unfortunately, the model does not correspond to inference in a GP model (see Section 2.3.1 in their paper). Our approach does not have this issue and we can express Gaussian posterior approximations on a Bayesian DNN as inference in a GP regression model.

## 2 Deep Neural Networks (DNNs) and Gaussian Processes (GPs)

The goal of this paper is to present a theoretical relationship between training methods of DNNs and GPs. DNNs are typically trained by minimizing an empirical loss between the data and the predictions. For example, in supervised learning with a dataset $\mathcal{D} := \{(\mathbf{x}_i, \mathbf{y}_i)\}_{i=1}^N$ of $N$ examples of input $\mathbf{x}_i \in \mathbb{R}^D$ and output $\mathbf{y}_i \in \mathbb{R}^K$, we can minimize a loss of the following form:

$$\bar{\ell}(\mathcal{D}, \mathbf{w}) := \sum_{i=1}^N \ell_i(\mathbf{w}) + \tfrac{1}{2}\delta\mathbf{w}^\top\mathbf{w}, \quad \text{where } \ell_i(\mathbf{w}) := \ell(\mathbf{y}_i, \mathbf{f}_w(\mathbf{x}_i)), \tag{1}$$

where $\mathbf{f}_w(\mathbf{x}) \in \mathbb{R}^K$ denotes the DNN outputs with weights $\mathbf{w} \in \mathbb{R}^P$, $\ell(\mathbf{y}, \mathbf{f}(\mathbf{x}))$ denotes a loss function between an output $\mathbf{y}$ and the function $\mathbf{f}(\mathbf{x})$, and $\delta$ is a small $L_2$ regularizer.[2] We assume the loss function to be twice differentiable and strictly convex in $\mathbf{f}$ (e.g., squared loss and cross-entropy loss). An attractive feature of DNNs is that they can be trained using stochastic-gradient (SG) methods [11]. Such methods scale well to large data settings.

GP models use an entirely different modeling approach which is based on directly modeling the functions rather than the parameters. For example, for regression problems with scalar outputs $y_i \in \mathbb{R}$, consider the following linear *basis-function* model with a nonlinear feature-map $\phi(\mathbf{x}) : \mathbb{R}^D \mapsto \mathbb{R}^P$:

$$y = \phi(\mathbf{x})^\top\mathbf{w} + \epsilon, \quad \text{with } \epsilon \sim \mathcal{N}(0, \sigma^2), \quad \text{and } \mathbf{w} \sim \mathcal{N}(0, \delta^{-1}\mathbf{I}_P), \tag{2}$$

where $\mathbf{I}_P$ is a $P \times P$ identity matrix and $\sigma^2$ is the output noise variance. Defining the function to be $f(\mathbf{x}) := \phi(\mathbf{x})^\top\mathbf{w}$, the predictive distribution $p(f(\mathbf{x}_*)|\mathbf{x}_*, \mathcal{D})$ at a new test input $\mathbf{x}_*$ is equal to that of the following model directly defined with a GP prior over $f(\mathbf{x})$ [23]:

$$y = f(\mathbf{x}) + \epsilon, \quad \text{with } f(\mathbf{x}) \sim \mathcal{GP}\left(0, \kappa(\mathbf{x}, \mathbf{x}')\right), \tag{3}$$

where $\kappa(\mathbf{x}, \mathbf{x}') := \mathbb{E}[f(\mathbf{x})f(\mathbf{x}')] = \delta^{-1}\phi(\mathbf{x})^\top\phi(\mathbf{x}')$ is the *covariance function* or *kernel* of the GP. The function-space model is more general in the sense that it can also deal with infinite-dimensional vector feature maps $\phi(\mathbf{x})$, giving us a *nonparametric* model. This view has been used to show that as a DNN becomes infinitely wide it tends to a GP, by essentially showing that averaging over $p(\mathbf{w})$ with the feature map induced by a DNN leads to a GP covariance function [16].

An attractive property of the *function-space* formulation as opposed to the *weight-space* formulation, such as (1), is that the posterior distribution has a closed-form expression. Another attractive property is that the posterior is usually unimodal, unlike the loss $\bar{l}(\mathcal{D}, \mathbf{w})$ which is typically nonconvex. Unfortunately, the computation of the posterior takes $O(N^3)$ which is infeasible for large datasets. GPs also require choosing a good kernel [23]. Unlike DNNs, inference in GPs remains much more difficult.

To summarize, despite the similarities between the two models, their training methods are fundamentally different. While DNNs employ stochastic optimization, GPs use closed-form updates. How can we relate these seemingly different training procedures in practical settings, e.g., without assuming infinite-width DNNs? In this paper, we provide an answer to this question. We derive theoretical results that relate the solutions and iterations of deep-learning algorithms to GP inference. We do so by first finding a Gaussian posterior approximation (Step A in Fig. 1), then use it to find a linear basis-function model (Step B in Fig. 1) and its corresponding GP (Step C in Fig. 1). We start in the next section with our first theoretical result.

# 3 Relating Minima of the Loss to GP Inference via Laplace Approximation

In this section, we present theoretical results relating minima of a deep-learning loss (1) to inference in GP models. A local minimizer $\mathbf{w}_*$ of the loss (1) satisfies the following first-order and second-order conditions [17]: $\nabla_w \bar{\ell}(\mathcal{D}, \mathbf{w}_*) = 0$ and $\nabla^2_{ww} \bar{\ell}(\mathcal{D}, \mathbf{w}_*) \succ 0$. Deep-learning optimizers, such as RMSprop and Adam, aim to find such minimizers, and our goal is to relate them to GP inference.

**Step A (Laplace Approximation):** To do so, we will use an approximate inference method called the Laplace approximation [1]. The minima of the loss (1) corresponds to a mode of the Bayesian model: $p(\mathcal{D}, \mathbf{w}) := \prod_{i=1}^{N} e^{-\ell_i(\mathbf{w})} p(\mathbf{w})$ with prior distribution $p(\mathbf{w}) := \mathcal{N}(\mathbf{w}|0, \delta^{-1}\mathbf{I}_P)$, assuming that the posterior is well-defined. The posterior distribution $p(\mathbf{w}|\mathcal{D}) = p(\mathcal{D}, \mathbf{w})/p(\mathcal{D})$ is usually computationally intractable and requires computationally-feasible approximation methods. The Laplace approximation uses the following Gaussian approximation for the posterior:

$$p(\mathbf{w}|\mathcal{D}) \approx \mathcal{N}(\mathbf{w}|\boldsymbol{\mu}, \boldsymbol{\Sigma}), \text{ where } \boldsymbol{\mu} = \mathbf{w}_* \text{ and } \boldsymbol{\Sigma}^{-1} = \sum_{i=1}^{N} \nabla^2_{ww} \ell_i(\mathbf{w}_*) + \delta\mathbf{I}_P. \quad (4)$$

This approximation can be directly built using the solutions found by deep-learning optimizers.

**Step B (Linear Model):** The next step is to find a linear basis-function model whose posterior distribution is equal to the Gaussian approximation (4). We will now show that this is always possible whenever the gradient and Hessian of the loss[3] can be approximated as follows:

$$\nabla_w \ell(\mathbf{w}) \approx \boldsymbol{\phi}_w(\mathbf{x})\mathbf{v}_w(\mathbf{x}, \mathbf{y}), \qquad \nabla^2_{ww} \ell(\mathbf{w}) \approx \boldsymbol{\phi}_w(\mathbf{x})\mathbf{D}_w(\mathbf{x}, \mathbf{y})\boldsymbol{\phi}_w(\mathbf{x})^\top, \quad (5)$$

where $\boldsymbol{\phi}_w(\mathbf{x})$ is a $P \times Q$ feature matrix with $Q$ as a positive integer, $\mathbf{v}_w(\mathbf{x}, \mathbf{y})$ is a $Q$ length vector, and $\mathbf{D}_w(\mathbf{x}, \mathbf{y})$ is a $Q \times Q$ symmetric positive-definite matrix. We will now present results for a specific choice $\boldsymbol{\phi}_w, \mathbf{v}_w$, and $\mathbf{D}_w$. Our proof trivially generalizes to arbitrary choices of these quantities.

For the loss of form (1), the gradient and Hessian take the following form [15, 17]:

$$\nabla_w \ell(\mathbf{w}) = \mathbf{J}_w(\mathbf{x})^\top \mathbf{r}_w(\mathbf{x}, \mathbf{y}), \quad \nabla^2_{ww} \ell(\mathbf{w}) = \mathbf{J}_w(\mathbf{x})^\top \boldsymbol{\Lambda}_w(\mathbf{x}, \mathbf{y})\mathbf{J}_w(\mathbf{x}) + \mathbf{H}_f \mathbf{r}_w(\mathbf{x}, \mathbf{y}), \quad (6)$$

where $\mathbf{J}_w(\mathbf{x}) := \nabla_w \mathbf{f}_w(\mathbf{x})^\top$ is a $K \times P$ Jacobian matrix, $\mathbf{r}_w(\mathbf{x}, \mathbf{y}) := \nabla_f \ell(\mathbf{y}, \mathbf{f})$ is the *residual* vector evaluated at $\mathbf{f} := \mathbf{f}_w(\mathbf{x})$, $\boldsymbol{\Lambda}_w(\mathbf{x}, \mathbf{y}) := \nabla^2_{ff} \ell(\mathbf{y}, \mathbf{f})$, referred to as the *noise precision*, is the $K \times K$ Hessian matrix of the loss evaluated at $\mathbf{f} := \mathbf{f}_w(\mathbf{x})$, and $\mathbf{H}_f := \nabla^2_{ww} \mathbf{f}_w(\mathbf{x})$. The similarity between (5) and (6) is striking. In fact, if we ignore the second term for the Hessian $\nabla^2_{ww} \ell(\mathbf{w})$ in (6), we get the well-known *Generalized Gauss-Newton* (GGN) approximation [15, 17]:

$$\nabla^2_{ww} \ell(\mathbf{w}) \approx \mathbf{J}_w(\mathbf{x})^\top \boldsymbol{\Lambda}_w(\mathbf{x}, \mathbf{y})\mathbf{J}_w(\mathbf{x}). \quad (7)$$

This gives us one choice for the approximation (5) where we can set $\boldsymbol{\phi}_w(\mathbf{x}) := \mathbf{J}_w(\mathbf{x})^\top, \mathbf{v}_w(\mathbf{x}, \mathbf{y}) := \mathbf{r}_w(\mathbf{x}, \mathbf{y})$, and $\mathbf{D}_w(\mathbf{x}, \mathbf{y}) := \boldsymbol{\Lambda}_w(\mathbf{x}, \mathbf{y})$.

We are now ready to present our first theoretical result. Consider a Laplace approximation (4) but with the GGN approximation (7) for the Hessian. We refer to this as *Laplace-GGN*, and denote it by $\mathcal{N}(\mathbf{w}|\boldsymbol{\mu}, \widetilde{\boldsymbol{\Sigma}})$ where $\widetilde{\boldsymbol{\Sigma}}$ is the covariance obtained by using the GGN approximation. We denote the Jacobian, noise-precision, and residual at $\mathbf{w} = \mathbf{w}_*$ by $\mathbf{J}_*(\mathbf{x}), \boldsymbol{\Lambda}_*(\mathbf{x}, \mathbf{y})$, and $\mathbf{r}_*(\mathbf{x}, \mathbf{y})$. We construct a transformed dataset $\widetilde{\mathcal{D}} = \{(\mathbf{x}_i, \tilde{\mathbf{y}}_i)\}_{i=1}^{N}$ where the outputs $\tilde{\mathbf{y}}_i \in \mathbb{R}^K$ are equal to $\tilde{\mathbf{y}}_i := \mathbf{J}_*(\mathbf{x}_i)\mathbf{w}_* - \boldsymbol{\Lambda}_*(\mathbf{x}_i, \mathbf{y}_i)^{-1}\mathbf{r}_*(\mathbf{x}_i, \mathbf{y}_i)$. We consider the following linear model for $\widetilde{\mathcal{D}}$:

$$\tilde{\mathbf{y}} = \mathbf{J}_*(\mathbf{x})\mathbf{w} + \boldsymbol{\epsilon}, \text{ with } \boldsymbol{\epsilon} \sim \mathcal{N}(0, (\boldsymbol{\Lambda}_*(\mathbf{x}, \mathbf{y}))^{-1}) \text{ and } \mathbf{w} \sim \mathcal{N}(0, \delta^{-1}\mathbf{I}_P). \quad (8)$$

The following theorem states our result.

**Theorem 1.** *The Laplace approximation $\mathcal{N}(\mathbf{w}|\boldsymbol{\mu}, \widetilde{\boldsymbol{\Sigma}})$ is equal to the posterior distribution $p(\mathbf{w}|\widetilde{\mathcal{D}})$ of the linear model* (8).

A proof is given in Appendix A.1. The linear model uses $\mathbf{J}_*(\mathbf{x})$ as the nonlinear feature map, and the noise prevision $\boldsymbol{\Lambda}_*(\mathbf{x}, \mathbf{y})$ is obtained using the Hessian of the loss evaluated at $\mathbf{f}_{w_*}(\mathbf{x})$. The model is constructed such that its posterior is equal to the Laplace approximation and it exploits the quadratic approximation at $\mathbf{w}_*$. We now describe the final step relating the linear model to GPs.

**Step C (GP Model):** To get a GP model, we use the equivalence between the weight-space view shown in (2) and the function-space view shown in (3). With this, we get the following GP regression model whose predictive distribution $p(f(\mathbf{x}_*)|\mathbf{x}_*, \widetilde{\mathcal{D}})$ is equal to that of the linear model (8):

$$\tilde{\mathbf{y}} = \mathbf{f}(\mathbf{x}) + \boldsymbol{\epsilon}, \quad \text{with } \mathbf{f}(\mathbf{x}) \sim \mathcal{GP}\left(0, \delta^{-1}\mathbf{J}_*(\mathbf{x})\mathbf{J}_*(\mathbf{x}')^\top\right). \tag{9}$$

Note that the kernel here is a *multi-dimensional* $K \times K$ kernel. The steps A, B, and C together convert a DNN defined in the weight-space to a GP defined in the function-space. We refer to this approach as "DNN2GP".

The resulting GP predicts in the space of outputs $\tilde{\mathbf{y}}$ and therefore results in different predictions than the DNN, but it is connected to it through the Laplace approximation as shown in Theorem 1. In Appendix B, we describe prediction of the outputs $\mathbf{y}$ (instead of $\tilde{\mathbf{y}}$) using this GP. Note that our approach leads to a heteroscedastic GP which could be beneficial. Even though our derivation assumes a Gaussian prior and DNN model, the approach holds for other types of priors and models.

**Relationship to NTK:** The GP kernel in (9) is the Neural Tangent Kernel [4] (NTK) [8] which has desirable theoretical properties. As the width of the DNN is increasing to infinity, the kernel converges in probability to a deterministic kernel and also remains asymptotically constant during training. Our kernel is the NTK defined at $\mathbf{w}_*$ and is expected to have similar properties. It is also likely that, as the DNN width is increased, the Laplace-GGN approximation has similar properties as a GP posterior, and can be potentially used to improve the performance of DNNs. For example, we can use GPs to tune hyperparameters of DNNs. The function-space view is also useful to understand relationships between data examples. Another advantage of our approach is that we can derive kernels other than the NTK. Any approximation of the form (5) will always result in a linear model similar to (8).

**Accuracy of the GGN approximation:** This approximation is accurate when the model $\mathbf{f}_w(\mathbf{x})$ can fit the data well, in which case the residuals $\mathbf{r}_w(\mathbf{x}, \mathbf{y})$ are close to zero for all training examples and the second term in (6) goes to zero [2, 15, 17]. The GGN approximation is a convenient option to derive DNN2GP, but, as it is clear from (5), other types of approximations can also be used.

# 4 Relating Iterations of a Deep-Learning Algorithm to GP Inference via VI

In this section, we present theoretical results relating iterations of an RMSprop-like algorithm to GP inference. The RMSprop algorithm [21] uses the following updates (all operations are element-wise):

$$\mathbf{w}_{t+1} \leftarrow \mathbf{w}_t - \alpha_t \left(\sqrt{\mathbf{s}_{t+1}} + \Delta\right)^{-1} \hat{\mathbf{g}}(\mathbf{w}_t), \qquad \mathbf{s}_{t+1} \leftarrow (1 - \beta_t)\mathbf{s}_t + \beta_t \left(\hat{\mathbf{g}}(\mathbf{w}_t)\right)^2, \tag{10}$$

where $t$ is the iteration, $\alpha_t > 0$ and $0 < \beta_t < 1$ are learning rates, $\Delta > 0$ is a small scalar, and $\hat{\mathbf{g}}(\mathbf{w})$ is a stochastic-gradient estimate for $\bar{\ell}(\mathcal{D}, \mathbf{w})$ obtained using minibatches. Our goal is to relate the iterates $\mathbf{w}_t$ to GP inference using our DNN2GP approach, but this requires a posterior approximation defined at each $\mathbf{w}_t$. We cannot use the Laplace approximation because it is only valid at $\mathbf{w}_*$. We will instead use a version of RMSprop proposed in [10] for variational inference (VI), which enables us to construct a GP inference problem at each $\mathbf{w}_t$.

**Step A (Variational Inference):** The variational online-Newton (VON) algorithm proposed in [10] optimizes the variational objective, but takes an algorithmic form similar to RMSprop (see a detailed discussion in [10]). Below, we show a batch version of VON, derived using Eq. (54) in [10]:

$$\boldsymbol{\mu}_{t+1} \leftarrow \boldsymbol{\mu}_t - \beta_t(\mathbf{S}_{t+1} + \delta\mathbf{I}_P)^{-1}\mathbb{E}_{q_t(w)}\left[\nabla_w\bar{\ell}(\mathcal{D}, \mathbf{w})\right], \tag{11}$$

$$\mathbf{S}_{t+1} \leftarrow (1 - \beta_t)\mathbf{S}_t + \beta_t \sum_{i=1}^N \mathbb{E}_{q_t(w)}\left[\nabla_{ww}^2\ell_i(\mathbf{w})\right], \tag{12}$$

where $\mathbf{S}_t$ is a scaling matrix similar to the scaling vector $\mathbf{s}_t$ in RMSprop, and the Gaussian approximation at iteration $t$ is defined as $q_t(\mathbf{w}) := \mathcal{N}(\mathbf{w}|\boldsymbol{\mu}_t, \boldsymbol{\Sigma}_t)$ where $\boldsymbol{\Sigma}_t := (\mathbf{S}_t + \delta\mathbf{I}_P)^{-1}$. Since there are no closed-form expressions for the expectations, the Monte Carlo (MC) approximation is used.

**Step B (Linear Model):** As before, we assume the choices for (5) obtained by using the GGN approximation (7). We consider the variant for VON where the GGN approximation is used for the Hessian and MC approximation is used for the expectations with respect to $q_t(\mathbf{w})$. We call this the

Variational Online GGN or VOGGN algorithm. A similar algorithm has recently been used in [19] where it shows competitive performance to Adam and SGD.

We now present a theorem relating iterations of VOGGN to linear models. We denote the Gaussian approximation obtained at iteration $t$ by $\tilde{q}_t(\mathbf{w}) := \mathcal{N}(\mathbf{w}|\boldsymbol{\mu}_t, \widetilde{\boldsymbol{\Sigma}}_t)$ where $\widetilde{\boldsymbol{\Sigma}}_t$ is used to emphasize the GGN approximation. We present theoretical results for VOGGN with 1 MC sample which is denoted by $\mathbf{w}_t \sim \tilde{q}_t(\mathbf{w})$. Our proof in Appendix A.2 discusses a more general setting with multiple MC samples. Similarly to the previous section, we first define a transformed dataset: $\widetilde{\mathcal{D}}_t := \{(\mathbf{x}_i, \tilde{\mathbf{y}}_{i,t})\}_{i=1}^N$ where $\tilde{\mathbf{y}}_{i,t} := \mathbf{J}_{w_t}(\mathbf{x}_i)\mathbf{w}_t - \boldsymbol{\Lambda}_{w_t}(\mathbf{x}_i, \mathbf{y}_i)^{-1}\mathbf{r}_{w_t}(\mathbf{x}_i, \mathbf{y}_i)$, and then a linear basis-function model:

$$\tilde{\mathbf{y}}_t = \mathbf{J}_{w_t}(\mathbf{x})\mathbf{w} + \boldsymbol{\epsilon}, \text{ with } \boldsymbol{\epsilon} \sim \mathcal{N}(0, (\beta_t \boldsymbol{\Lambda}_{w_t}(\mathbf{x}, \mathbf{y}))^{-1}) \text{ and } \mathbf{w} \sim \mathcal{N}(\mathbf{m}_t, \mathbf{V}_t) \qquad (13)$$

with $\mathbf{V}_t^{-1} := (1 - \beta_t)\widetilde{\boldsymbol{\Sigma}}_t^{-1} + \beta_t \delta \mathbf{I}_P$ and $\mathbf{m}_t := (1 - \beta_t)\mathbf{V}_t \widetilde{\boldsymbol{\Sigma}}_t^{-1}\mathbf{w}_t$. The model is very similar to the one obtained for Laplace approximation, but is now defined using the iterates $\mathbf{w}_t$ instead of the minimum $\mathbf{w}_*$. The prior over $\mathbf{w}$ is not the standard Gaussian anymore, rather a correlated Gaussian derived from $q_t(\mathbf{w})$. The theorem below states the result (a proof is given in Appendix A.2).

**Theorem 2.** *The Gaussian approximation $\mathcal{N}(\mathbf{w}|\mathbf{w}_{t+1}, \widetilde{\boldsymbol{\Sigma}}_{t+1})$ at iteration $t + 1$ of the VOGGN update is equal to the posterior distribution $p(\mathbf{w}|\widetilde{\mathcal{D}}_t)$ of the linear model* (13).

**Step C (GP Model):** The linear model (13) has the same predictive distribution as the GP below:

$$\tilde{\mathbf{y}}_t = \mathbf{f}_t(\mathbf{x}) + \boldsymbol{\epsilon}, \quad \text{with } \mathbf{f}_t(\mathbf{x}) \sim \mathcal{GP}\left(\mathbf{J}_{w_t}(\mathbf{x})\mathbf{m}_t, \mathbf{J}_{w_t}(\mathbf{x})\mathbf{V}_t \mathbf{J}_{w_t}(\mathbf{x}')^\top\right). \qquad (14)$$

The kernel here is similar to the NTK but now there is a covariance term $\mathbf{V}_t$ which incorporates the effect of the previous $q_t(\mathbf{w})$ as a prior. Our DNN2GP approach shows that one iteration of VOGGN in the weight-space is equivalent to inference in a GP regression model defined in a transformed function-space with respect to a kernel similar to the NTK. This can be compared with the results in [8], where learning by plain gradient descent is shown to be equivalent to kernel gradient descent in function-space. Similarly to the Laplace case, the resulting GP predicts in the space of outputs $\tilde{\mathbf{y}}_t$, but predictions for $\mathbf{y}_t$ can be obtained using a method described in Appendix B.

**A Deep-Learning Optimizer Derived from VOGGN:** The VON algorithm, even though similar to RMSprop, does not converge to the minimum of the loss. This is because it optimizes the variational objective. Fortunately, a slight modification of this algorithm gives us a deep-learning optimizer which is similar to RMSprop but is guaranteed to converge to the minimum of the loss. For this, we approximate the expectations in the updates (11)-(12) at the mean $\boldsymbol{\mu}_t$. This is called the *zeroth-order delta approximation*; see Appendix A.6 in [9] for details of this method. Using this approximation and denoting the mean $\boldsymbol{\mu}_t$ by $\mathbf{w}_t$, we get the following update:

$$\mathbf{w}_{t+1} \leftarrow \mathbf{w}_t - \beta_t(\hat{\mathbf{S}}_{t+1} + \delta \mathbf{I}_P)^{-1}\nabla_w \bar{\ell}(\mathcal{D}, \mathbf{w}_t), \qquad \hat{\mathbf{S}}_{t+1} \leftarrow (1 - \beta_t)\hat{\mathbf{S}}_t + \beta_t \sum_{i=1}^N \left[\nabla_{ww}^2 \ell_i(\mathbf{w}_t)\right].$$

We refer to this as Online GGN or OGGN method. A fixed point $\mathbf{w}_*$ of this iteration is also a minimizer of the loss since we have $\nabla_w \bar{\ell}(\mathcal{D}, \mathbf{w}_*) = 0$. Unlike RMSprop, at each iteration, we still get a Gaussian approximation $\hat{q}_t(\mathbf{w}) := \mathcal{N}(\mathbf{w}|\mathbf{w}_t, \hat{\boldsymbol{\Sigma}}_t)$ with $\hat{\boldsymbol{\Sigma}}_t := (\hat{\mathbf{S}}_t + \delta \mathbf{I}_P)^{-1}$. Therefore, the posterior of the linear model from Theorem (2) is equivalent to $\hat{q}_t$ when $\widetilde{\boldsymbol{\Sigma}}_t$ is replaced by $\hat{\boldsymbol{\Sigma}}_t$ (see Appendix A.3). In conclusion, by using VI in our DNN2GP approach, we are able to relate the iterations of a deep-learning optimizer to GP inference.

**Implementation of DNN2GP:** In practice, both VOGGN and OGGN are computationally more expensive than RMSprop because they involve computation of full covariance matrices. To address this issue, we simply use the diagonal versions of these algorithms discussed in [10, 19]. Specifically, we use the VOGN and OGN algorithms discussed in [19]. This implies that $\mathbf{V}_t$ is a diagonal matrix and the GP kernel can be obtained without requiring any computation of large matrices. Only Jacobian computations are required. In our experiments, we also resort to computing the kernel over a subset of data instead of the whole data, which further reduces the cost.

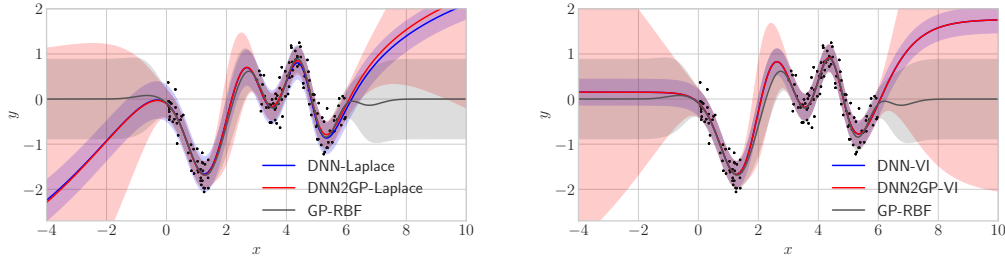

Figure 3: This figure shows a visualization of the predictive distributions on a modified version of the Snelson dataset [20]. The left figure shows Laplace and the right one shows VI. DNN2GP is our proposed method, elaborated upon in Appendix B, while DNN refers to a diagonal Gaussian approximation. We also compare to a GP with RBF kernel (GP-RBF). An MLP is used for DNN2GP and DNN. We see that, wherever the data is missing, the uncertainties are larger for our method than the others. For classification, we give an example in Fig. 9 in the appendix.

## 5    Experimental Results

### 5.1    Comparison of DNN2GP Uncertainty

In this section, we visualize the quality of the uncertainty of the GP obtained with our DNN2GP approach on a simple regression task. To approximate predicitive uncertainty for our approach, we use the method described in Appendix B. We use both Laplace and VI approximations, referred to as 'DNN2GP-Laplace' and 'DNN2GP-VI', respectively. We compare it to the uncertainty obtained using an MC approximation in the DNN (referred to as 'DNN-Laplace' and 'DNN-VI'). We also compare to a standard GP regression model with an RBF kernel (refer to as 'GP-RBF'), whose kernel hyperparameters are chosen by optimizing the GP marginal likelihood.

We consider a version of the Snelson dataset [20] where, to assess the 'in-between' uncertainty, we remove the data points between $x = 1.5$ and $x = 3$. We use a single hidden-layer MLP with 32 units and sigmoidal transfer function. Fig. 3 shows the results for Laplace (left) and VI (right) approximation. For Laplace, we use Adam [11], and, for VI, we use VOGN [10]. The uncertainty provided by DNN2GP is bigger than the other methods wherever the data is not observed.

### 5.2    GP Kernel and Predictive Distribution for Classification Datasets

In this section, we visualize the GP kernel and predictive distribution for DNNs trained on CIFAR-10 and MNIST. Our goal is to show that our GP kernel and its predictions enhance our understanding of a DNN's performance on classification tasks. We consider LeNet-5 [12] and compute both the Laplace and VI approximations. We show the visualization at the posterior mean.

The $K \times K$ GP kernel $\boldsymbol{\kappa}_*(\mathbf{x}, \mathbf{x}') := \mathbf{J}_*(\mathbf{x})\mathbf{J}_*(\mathbf{x}')^\top$ results in a kernel matrix of dimensionality $NK \times NK$ which makes it difficult to visualize for our datasets. To simplify, we compute the sum of the diagonal entries of $\boldsymbol{\kappa}_*(\mathbf{x}, \mathbf{x}')$ to get an $N \times N$ matrix. This corresponds to modelling the output for each class with an individual GP and then summing the kernels of these GPs. We also visualize the GP posterior mean: $\mathbb{E}[\mathbf{f}(\mathbf{x})|\mathcal{D}] = \mathbb{E}[\mathbf{J}_*(\mathbf{x})\mathbf{w}|\mathcal{D}] = \mathbf{J}_*(\mathbf{x})\mathbf{w}_* \in \mathbb{R}^K$. and use the reparameterization that allows to predict in the data space $\mathbf{y}$ instead of $\tilde{\mathbf{y}}$ which is explained in Appendix B.

Fig. 4a shows the GP kernel matrix and the posterior mean for the Laplace approximation on MNIST. The rows and columns containing 300 data examples are grouped according to the classes. The kernel matrix clearly shows the correlations learned by the DNN. As expected, each row in the posterior mean also reflects that the classes are correctly classified (DNN test accuracy is 99%). Fig. 4b shows the GP posterior mean after reparameterization for CIFAR-10 where we see a more noisy pattern due to a lower accuracy of around 68% on this task.

Fig. 4d shows the two components of the predictive variances that can be interpreted as "aleatoric" and "epistemic" uncertainty. As shown in Eq. (48) in Appendix B.2, for a multiclass classification loss, the variance of the prediction of a label at an input $\mathbf{x}_*$ is equal to $\boldsymbol{\Lambda}_*(\mathbf{x}_*) + \boldsymbol{\Lambda}_*(\mathbf{x}_*)\mathbf{J}_*(\mathbf{x}_*)\widetilde{\boldsymbol{\Sigma}}\mathbf{J}_*(\mathbf{x}_*)^\top\boldsymbol{\Lambda}_*(\mathbf{x}_*)$. Similar to the linear basis function model, the two terms here have an interpretation (e.g., see Eq. 3.59 in [1]). The first term can be interpreted as the aleatoric uncertainty (label noise), while the second term takes a form that resembles the epistemic uncertainty

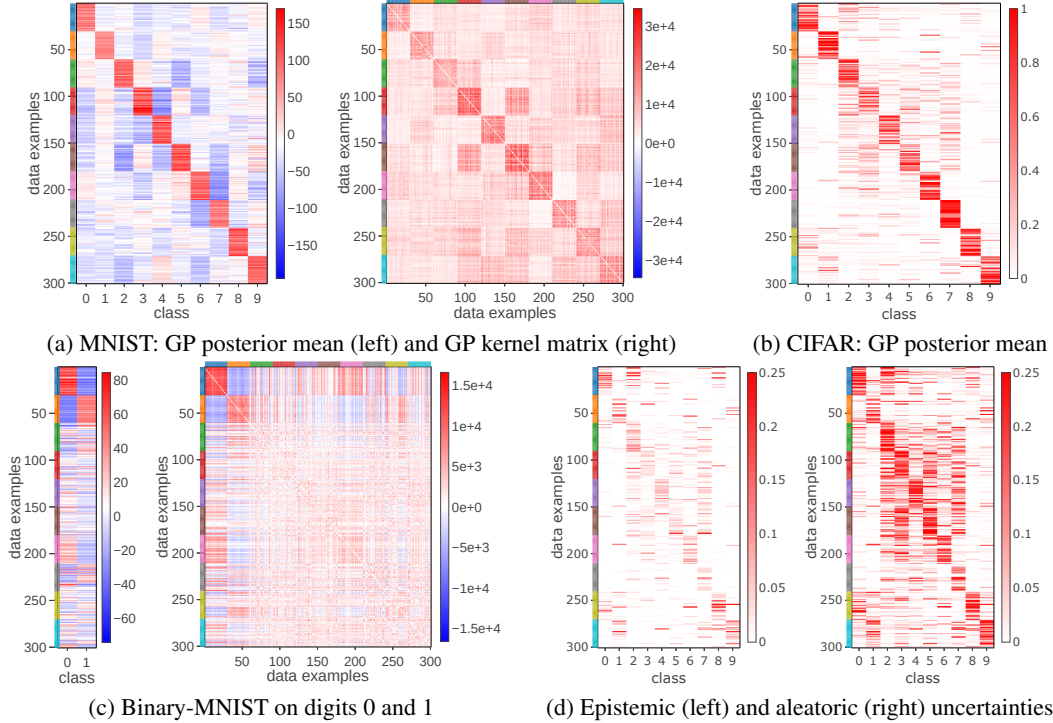

(a) MNIST: GP posterior mean (left) and GP kernel matrix (right)　　(b) CIFAR: GP posterior mean

(c) Binary-MNIST on digits 0 and 1　　(d) Epistemic (left) and aleatoric (right) uncertainties

Figure 4: DNN2GP kernels, posterior means and uncertainties with LeNet5 of 300 samples on binary MNIST in Fig. (c), MNIST in Fig. (a), and CIFAR-10 in Fig. (b,d). The colored regions on the y-axis mark the classes. Fig. (a) shows the kernel and the predictive mean for the Laplace approximation, which gives 99% test accuracy. We see in the kernel that examples with same class labels are correlated. Fig. (c) shows the same for binary MNIST trained only on digits 0 and 1 by using VI. The kernel clearly shows the out-of-class predictive behavior where predictions are not certain. Fig. (b) and (d) show the Laplace-GP on the more complex CIFAR-10 data set where we obtain 68% accuracy. Fig. (d) shows the two components of the predictive variance for CIFAR-10 that can be interpreted as epistemic (left) and aleatoric (right) uncertainties. The estimated epistemic uncertainty is much lower than the aleatoric uncertainty, implying that the model is not flexible enough. This is plausible since the accuracy of the model is not too high (merely 68%).

(model noise). Fig. 4d shows these for CIFAR-10 where we see that the uncertainty of the model is low (left) and the label noise rather high (right). This interpretation implies that the model is unable to flexibly model the data and instead explains it with high label noise.

In Fig. 4c, we study the kernel for classes *outside* of the training dataset using VI. We train LeNet-5 on digits 0 and 1 with VOGN and visualize the predictive mean and kernel on all 10 classes denoted by differently colored regions on the y-axis. We can see that there are slight correlations to the out-of-class samples but no overconfident predictions. In contrast, the pattern between 0 and 1 is quite strong. The kernel obtained with DNN2GP helps to interpret and visualize such correlations.

## 5.3 Tuning the Hyperparameters of a DNN Using the GP Marginal Likelihood

In this section, we demonstrate the tuning of DNN hyperparameters by using the GP marginal likelihood on a real and synthetic regression dataset. In the deep-learning literature, this is usually done using cross-validation. Our goal is to demonstrate that with DNN2GP we can do this by simply computing the marginal likelihood on the *training* set.

We generate a synthetic regression dataset ($N = 100$; see Fig. 5) where there are a few data points around $x = 0$ but plenty away from it. We fit the data by using a neural network with single hidden layer of 20 units and $\tanh$ nonlinearity. Our goal is to tune the regularization parameter $\delta$ to trade-off underfitting vs overfitting. Fig. 5b and 5c show the train log marginal-likelihood obtained with the GP obtained by DNN2GP, along with the test and train mean-square error (MSE) obtained using a point estimate. Black stars indicate the hyperparameters chosen by using the test loss and log marginal

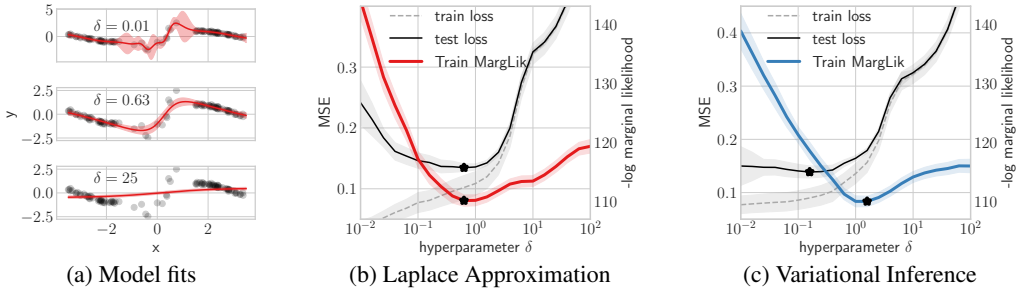

| (a) Model fits | (b) Laplace Approximation | (c) Variational Inference |

Figure 5: This figure demonstrates the use of the GP marginal likelihood to tune hyperparameters of a DNN. We tune the regularization parameter $\delta$ on a synthetic dataset shown in (a). Fig. (b) and (c) show train and test MSE along with log of the marginal likelihoods on training data obtained with Laplace and VI respectively. We show the standard error over 10 runs. The optimal hyperparameters according to test loss and marginal-likelihood (shown with black stars) match well.

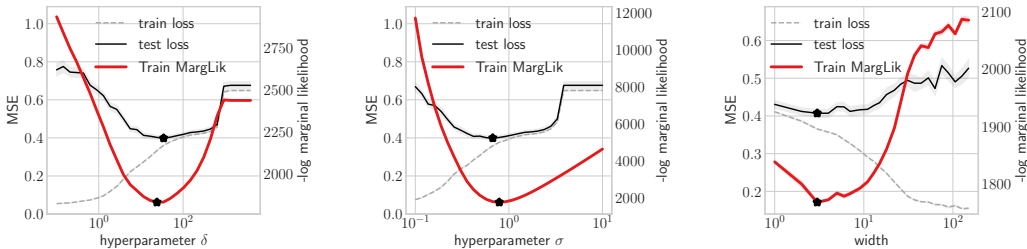

Figure 6: This is same as Fig. 5 but on a real dataset: UCI Red Wine Quality. All the plots use Laplace approximation, and the standard errors are estimated over 20 splits. We tune the following hyperparameters: the regularization parameter $\delta$ (left), the noise-variance $\sigma$ (middle), and the DNN width (right). The train log marginal-likelihood chooses hyperparameters that give a low test error.

likelihood, respectively. We clearly see that the train marginal-likelihood chooses hyperparameters that give low test error. The train MSE on the other hand overfits as $\delta$ is reduced.

Next, we discuss results for a real dataset: UCI Red Wine Quality ($N = 1599$) with an input-dimensionality of 12 and a scalar output. We use an MLP with 2 hidden layers 20 units each and $\tanh$ transfer function. We consider tuning the regularizer $\delta$, the noise-variance $\sigma$, and the DNN width. We use the Laplace approximation and tune one parameter at a time while keeping the others fixed (we use respectively $\sigma = 0.64$, $\delta = 30$ and $\sigma = 0.64$, $\delta = 3$, 1 hidden layer). Similarly to the synthetic data case, the train marginal-likelihood selects hyperparameters that give low test error. These experiments show that the DNN2GP framework can be useful to tune DNN hyperparameters, although this needs to be confirmed for larger networks than we used here.

# 6  Discussion and Future Work

In this paper, we present theoretical results connecting approximate inference on DNNs to GP posteriors. Our work enables the extraction of feature maps and GP kernels by simply training DNNs. It provides a natural way to combine the two different models.

Our hope is that our theoretical results will facilitate further research on combining strengths of DNNs and GPs. A computational bottleneck is the Jacobian computation which prohibits application to large problems. There are several ways to reduce this computation, e.g., by choosing a different type of GGN approximation that uses gradients instead of the Jacobians. Exploration of such methods is a future direction that needs to be pursued.

Exact inference on the GP model we derive is still computationally infeasible for large problems. However, further approximations could enable inference on bigger datasets. Finally, our work opens many other interesting avenues where a combination of GPs and DNNs can be useful such as model selection, deep reinforcement learning, Bayesian optimization, active learning, interpretation, etc. We hope that our work enables the community to conduct further research on such problems.

**Acknowledgements**

We would like to thank Kazuki Osawa (Tokyo Institute of Technology), Anirudh Jain (RIKEN), and Runa Eschenhagen (RIKEN) for their help with the experiments. We would also like to thank Matthias Bauer (DeepMind) for discussions and useful feedback. Many thanks to Roman Bachmann (RIKEN) for helping with the visualization in Fig. 1. We also thank Stephan Mandt (UCI) for suggesting the marginal likelihood experiment. We thank the reviewers and the area chair for their feedback as well. We are also thankful for the RAIDEN computing system and its support team at the RIKEN Center for Advanced Intelligence Project which we used extensively for our experiments.

## Footnotes

[2]We can assume that $\delta$ is small enough that it does not affect the DNN's generalization.

[3]For notational convenience, we sometime use $\ell(\mathbf{w})$ to denote $\ell(\mathbf{y}, \mathbf{f}_w(\mathbf{x}))$.

[4]The NTK corrsponds to $\delta = 1$ which implies a standard normal prior on weights.

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
