[Supplementary Material]

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

# A   Proofs

In this section, we prove the theorems presented in the main text.

## A.1   Proof of Theorem 1

We begin with the Gaussian approximation of the Laplace approximation. We will then express its natural parameters in terms of the gradient and Hessians. Application of the GGN approximation and some further manipulation will show that the distribution correspond to the posterior of a linear model.

We start with the Laplace approximation (4) and express its natural parameters in terms of the gradient and Hessians. We denote the natural-parameters of this Gaussian approximation $\mathcal{N}(\mathbf{w}|\boldsymbol{\mu}, \boldsymbol{\Sigma})$ by $\boldsymbol{\eta} := \{\boldsymbol{\Sigma}^{-1}\boldsymbol{\mu}, -\frac{1}{2}\boldsymbol{\Sigma}^{-1}\}$. In (4), the second natural parameter is set to the following which is written in terms of the Hessian:

$$-\tfrac{1}{2}\boldsymbol{\Sigma}^{-1} = -\tfrac{1}{2}\left[\sum_{i=1}^{N} \nabla_{ww}^2 \ell_i(\mathbf{w}_*) + \delta\mathbf{I}_P\right]. \tag{15}$$

We can also express the first natural parameter in terms of the gradient and Hessians as shown below. We use the first-order stationary condition, that is, $\nabla_w \bar{\ell}(\mathcal{D}, \mathbf{w}_*) = 0$. Adding $\boldsymbol{\Sigma}^{-1}\boldsymbol{\mu}$ to the both sides of this condition, we get the following:

$$\boldsymbol{\Sigma}^{-1}\boldsymbol{\mu} = -\nabla_w \bar{\ell}(\mathcal{D}, \mathbf{w}_*) + \boldsymbol{\Sigma}^{-1}\boldsymbol{\mu} \tag{16}$$

$$= -\sum_{i=1}^{N} \nabla_w \ell_i(\mathbf{w}_*) - \delta\mathbf{w}_* + \left[\sum_{i=1}^{N} \nabla_{ww}^2 \ell_i(\mathbf{w}_*) + \delta\mathbf{I}_P\right]\mathbf{w}_* \tag{17}$$

$$= \sum_{i=1}^{N} \left[-\nabla_w \ell_i(\mathbf{w}_*) + \nabla_{ww}^2 \ell_i(\mathbf{w}_*)\mathbf{w}_*\right], \tag{18}$$

where in the second step, we substitute $\boldsymbol{\mu}$ by $\mathbf{w}_*$ and also use (15). With this, both natural parameters are now expressed in terms of the gradient and Hessian.

We will now substitute these in the Laplace approximation, denoted by $q_L(\mathbf{w}) := \mathcal{N}(\mathbf{w}|\boldsymbol{\mu}, \boldsymbol{\Sigma})$. Using (15) and (18), we get the following expression:

$$q_L(\mathbf{w}) = \frac{1}{\sqrt{(2\pi)^P|\boldsymbol{\Sigma}|}} \exp\left[-\tfrac{1}{2}(\mathbf{w} - \boldsymbol{\mu})^\top \boldsymbol{\Sigma}^{-1}(\mathbf{w} - \boldsymbol{\mu})\right] \tag{19}$$

$$\propto \exp\left[-\tfrac{1}{2}\mathbf{w}^\top(\boldsymbol{\Sigma}^{-1})\mathbf{w} + \mathbf{w}^\top(\boldsymbol{\Sigma}^{-1}\boldsymbol{\mu})\right] \tag{20}$$

$$= \exp\left(\frac{-\delta\mathbf{w}^\top\mathbf{w}}{2}\right) \prod_{i=1}^{N} \exp\left[-\tfrac{1}{2}\mathbf{w}^\top \nabla_{ww}^2 \ell_i(\mathbf{w}_*)\mathbf{w} + \mathbf{w}^\top \left\{-\nabla_w \ell_i(\mathbf{w}_*) + \nabla_{ww}^2 \ell_i(\mathbf{w}_*)\mathbf{w}_*\right\}\right], \tag{21}$$

where in the last line we use (15) and (18).

Now, we will employ the GGN approximation (7) which gives us the Laplace-GGN approximation $\mathcal{N}(\mathbf{w}|\boldsymbol{\mu}, \widetilde{\boldsymbol{\Sigma}})$, shown below:

$$\exp\left(\frac{-\delta\mathbf{w}^\top\mathbf{w}}{2}\right) \prod_{i=1}^{N} \exp\left[\frac{-1}{2}\mathbf{w}^\top \mathbf{J}_*(\mathbf{x}_i)^\top \boldsymbol{\Lambda}_{i,*}\mathbf{J}_*(\mathbf{x}_i)\mathbf{w} + \mathbf{w}^\top \mathbf{J}_*(\mathbf{x}_i)^\top \left\{\boldsymbol{\Lambda}_{i,*}\mathbf{J}_*(\mathbf{x}_i)\mathbf{w}_* - \mathbf{r}_{i,*}\right\}\right], \tag{22}$$

where for notational convenience we have denoted $\boldsymbol{\Lambda}_{i,*} := \boldsymbol{\Lambda}_*(\mathbf{x}_i, \mathbf{y}_i)$ and $\mathbf{r}_{i,*} := \mathbf{r}_*(\mathbf{x}_i, \mathbf{y}_i)$.

A key point here is that each term in the product over $i$ in (22) is proportional to a Gaussian distribution, provided that $\boldsymbol{\Lambda}_{i,*} \succ 0$, which is the case since we assume the loss function to be strictly convex. We will now express each term in the product, as a likelihood over a *pseudo-output* defined as $\tilde{\mathbf{y}}_i := \mathbf{J}_*(\mathbf{x}_i)\mathbf{w}_* - \boldsymbol{\Lambda}_{i,*}^{-1}\mathbf{r}_{i,*}$. Using this and completing the square within each term in the product over $i$ in (22), we get the following:

$$\tilde{q}_L(\mathbf{w}) := \mathcal{N}(\mathbf{w}|\boldsymbol{\mu}, \widetilde{\boldsymbol{\Sigma}}) \propto \mathcal{N}(\mathbf{w}|0, \delta^{-1}\mathbf{I}_P) \prod_{i=1}^{N} \mathcal{N}(\tilde{\mathbf{y}}_i|\mathbf{J}_*(\mathbf{x}_i)\mathbf{w}, \boldsymbol{\Lambda}_{i,*}^{-1}). \tag{23}$$

The right hand side of the above equation is proportional to the posterior distribution $p(\mathbf{w}|\widetilde{\mathcal{D}})$ given a transformed dataset $\widetilde{\mathcal{D}} := \{(\mathbf{x}_i, \tilde{\mathbf{y}}_i)\}_{i=1}^N$ of a linear basis-function model $\tilde{\mathbf{y}} = \mathbf{J}_*(\mathbf{x})\mathbf{w} + \epsilon$ with Gaussian noise $\epsilon \sim \mathcal{N}(0, (\mathbf{\Lambda}_*(\mathbf{x}, \mathbf{y}))^{-1})$ and prior distribution $\mathbf{w} \sim \mathcal{N}(0, \delta^{-1}\mathbf{I}_P)$. This completes the proof.

It is easy to see that the same proof works when using the approximations shown in (5). In that case, only the steps from (22) need to be modified. The proof also holds when a prior other than Gaussian and a model other than DNN is used. □

## A.2 Proof of Theorem 2

Similarly to the previous section, we start by writing the Gaussian approximation for VON. We will express its natural parameters in terms of the gradient and Hessians. A GGN approximation and some manipulation will show that the distributions found by VON correspond to posteriors of linear models.

The Gaussian approximation at the $t$'th iteration of VON is denoted by $q_t(\mathbf{w}) := \mathcal{N}(\mathbf{w}|\boldsymbol{\mu}_t, \mathbf{\Sigma}_t)$ where $\mathbf{\Sigma}_t := (\mathbf{S}_t + \delta\mathbf{I}_P)^{-1}$ is obtained from $\mathbf{S}_t$. Using this, we can rewrite the updates (11) and (12) in terms of $\boldsymbol{\mu}_t$ and $\mathbf{\Sigma}_t^{-1}$ as follows

$$\boldsymbol{\mu}_{t+1} = \boldsymbol{\mu}_t - \beta_t \mathbf{\Sigma}_{t+1} \left[ \sum_{i=1}^N \mathbb{E}_{q_t(w)} \left[ \nabla_w \ell_i(\mathbf{w}) \right] + \delta\boldsymbol{\mu}_t \right], \tag{24}$$

$$\mathbf{\Sigma}_{t+1}^{-1} = (1 - \beta_t)\mathbf{\Sigma}_t^{-1} + \beta_t \left[ \sum_{i=1}^N \mathbb{E}_{q_t(w)} \left[ \nabla_{ww}^2 \ell_i(\mathbf{w}) \right] + \delta\mathbf{I}_P \right]. \tag{25}$$

It is again more convenient if we can have an update formula for the natural-parameters of the Gaussian distribution $\mathcal{N}(\mathbf{w}|\boldsymbol{\mu}_t, \mathbf{\Sigma}_t)$, denoted by $\boldsymbol{\eta}_t := \{\mathbf{\Sigma}_t^{-1}\boldsymbol{\mu}_t, -\frac{1}{2}\mathbf{\Sigma}_t^{-1}\}$. So we use similar techniques to find an update for $\boldsymbol{\eta}_t$. In addition, since there are no closed-form expressions for the expectations above, we use $S$ number of samples $\mathbf{w}_t^{(s)} \sim q_t(\mathbf{w})$, for $s = 1, 2, \ldots, S$, and use Monte Carlo (MC) approximation.

Given (25), the update corresponding to the second natural-parameter is obvious and given by

$$-\frac{1}{2}\mathbf{\Sigma}_{t+1}^{-1} = (1 - \beta_t) \left[ -\frac{1}{2}\mathbf{\Sigma}_t^{-1} \right] - \frac{1}{2}\beta_t \left[ \sum_{i=1}^N \mathbb{E}_{q_t(w)} \left[ \nabla_{ww}^2 \ell_i(\mathbf{w}) \right] + \delta\mathbf{I}_P \right] \tag{26}$$

$$\approx (1 - \beta_t) \left[ -\frac{1}{2}\mathbf{\Sigma}_t^{-1} \right] - \frac{1}{2}\beta_t \left[ \frac{1}{S} \sum_{i,s=1}^{N,S} \nabla_{ww}^2 \ell_i(\mathbf{w}_t^{(s)}) + \delta\mathbf{I}_P \right], \tag{27}$$

where we have used an MC approximation in the second step.

To write the update for the first natural-parameter, we multiply (24) by $\mathbf{\Sigma}_{t+1}^{-1}$ and get

$$\mathbf{\Sigma}_{t+1}^{-1}\boldsymbol{\mu}_{t+1} = \mathbf{\Sigma}_{t+1}^{-1}\boldsymbol{\mu}_t - \beta_t \left[ \sum_{i=1}^N \mathbb{E}_{q_t(w)} \left[ \nabla_w \ell_i(\mathbf{w}) \right] + \delta\boldsymbol{\mu}_t \right] \tag{28}$$

$$= (1 - \beta_t) \left[ \mathbf{\Sigma}_t^{-1}\boldsymbol{\mu}_t \right] + \beta_t \sum_{i=1}^N \left[ -\mathbb{E}_{q_t(w)} \left[ \nabla_w \ell_i(\mathbf{w}) \right] + \mathbb{E}_{q_t(w)} \left[ \nabla_{ww}^2 \ell_i(\mathbf{w}) \right] \boldsymbol{\mu}_t \right] \tag{29}$$

$$\approx (1 - \beta_t) \left[ \mathbf{\Sigma}_t^{-1}\boldsymbol{\mu}_t \right] + \frac{\beta_t}{S} \sum_{i,s=1}^{N,S} \left[ -\nabla_w \ell_i(\mathbf{w}_t^{(s)}) + \nabla_{ww}^2 \ell_i(\mathbf{w}_t^{(s)})\boldsymbol{\mu}_t \right], \tag{30}$$

where in the second step, we replaced $\mathbf{\Sigma}_{t+1}^{-1}$ in the first term by (25). The posterior approximation $q_{t+1}(\mathbf{w})$ at time $t + 1$ can be written in terms of natural parameters as shown below:

$$q_{t+1}(\mathbf{w}) = \frac{1}{\sqrt{(2\pi)^P|\mathbf{\Sigma}_{t+1}|}} \exp \left[ -\frac{1}{2}(\mathbf{w} - \boldsymbol{\mu}_{t+1})^\top \mathbf{\Sigma}_{t+1}^{-1}(\mathbf{w} - \boldsymbol{\mu}_{t+1}) \right] \tag{31}$$

$$\propto \exp \left[ -\frac{1}{2}\mathbf{w}^\top (\mathbf{\Sigma}_{t+1}^{-1})\mathbf{w} + \mathbf{w}^\top (\mathbf{\Sigma}_{t+1}^{-1}\boldsymbol{\mu}_{t+1}) \right]. \tag{32}$$

By substituting the natural parameters from (27) and (30), we get the following update for $q_{t+1}(\mathbf{w})$, expressed in terms of the MC samples:

$$q_{t+1}(\mathbf{w}) \propto p(\mathbf{w})^{\beta_t} q_t(\mathbf{w})^{1-\beta_t} \times$$
$$\prod_{i,s=1}^{N,S} \exp\left[ -\frac{\beta_t}{2S} \mathbf{w}^\top \nabla_{ww}^2 \ell_i(\mathbf{w}_t^{(s)}) \mathbf{w} + \frac{\beta_t \mathbf{w}_t^\top}{S} \left\{ -\nabla_w \ell_i(\mathbf{w}_t^{(s)}) + \nabla_{ww}^2 \ell_i(\mathbf{w}_t^{(s)}) \boldsymbol{\mu}_t \right\} \right],$$
$$(33)$$

where $p(\mathbf{w}) = \mathcal{N}(\mathbf{w}|0, \delta^{-1}\mathbf{I}_P)$ is the prior distribution. For the product of posterior approximation at time $t$ and prior in (33), we obtain the following unnormalized Gaussian

$$p(\mathbf{w})^{\beta_t} q_t(\mathbf{w})^{1-\beta_t} = \mathcal{N}\left(\mathbf{w}|0, \delta^{-1}\mathbf{I}_P\right)^{\beta_t} \mathcal{N}\left(\mathbf{w}|\boldsymbol{\mu}_t, \boldsymbol{\Sigma}_t\right)^{1-\beta_t} \propto \mathcal{N}\left(\mathbf{w}|\mathbf{m}_t, \mathbf{V}_t\right), \qquad (34)$$

where $\mathbf{V}_t$ and $\mathbf{m}_t$ are given by

$$\mathbf{V}_t^{-1} := (1 - \beta_t)\boldsymbol{\Sigma}_t^{-1} + \beta_t \delta \mathbf{I}_P, \quad \mathbf{m}_t := (1 - \beta)\mathbf{V}_t \boldsymbol{\Sigma}_t^{-1} \boldsymbol{\mu}_t. \qquad (35)$$

Next, for the product over $i$ and $s$ in (33), we employ the GGN approximation (7) and get

$$\tilde{q}_{t+1}(\mathbf{w}) \propto \mathcal{N}\left(\mathbf{w}|\mathbf{m}_t, \mathbf{V}_t\right) \times$$
$$\prod_{i,s=1}^{N,S} \exp\left[ -\mathbf{w}^\top \mathbf{J}_{s,t}(\mathbf{x}_i)^\top \frac{\beta_t \boldsymbol{\Lambda}_{i,s,t}}{2S} \mathbf{J}_{s,t}(\mathbf{x}_i)\mathbf{w} + \frac{\beta_t \mathbf{w}^\top \mathbf{J}_{s,t}(\mathbf{x}_i)^\top}{S} \left\{ \boldsymbol{\Lambda}_{i,s,t}\mathbf{J}_{s,t}(\mathbf{x}_i)\boldsymbol{\mu}_t - \mathbf{r}_{i,s,t} \right\} \right],$$
$$(36)$$

where we have defined $\mathbf{J}_{s,t}(\mathbf{x}_i) := \mathbf{J}_{w_t^{(s)}}(\mathbf{x}_i)$, $\mathbf{r}_{i,s,t} := \mathbf{r}_{w_t^{(s)}}(\mathbf{x}_i, \mathbf{y}_i)$, and $\boldsymbol{\Lambda}_{i,s,t} := \boldsymbol{\Lambda}_{w_t^{(s)}}(\mathbf{x}_i, \mathbf{y}_i)$. The notation $\tilde{q}_{t+1}(\mathbf{w})$ is used to emphasize that GGN approximation is used in this update.

We are now ready to express each term in the product above as a Gaussian distribution. First, we define three quantities: $\mathbf{J}_t(\mathbf{x}), \mathbf{r}_t(\mathbf{x}, \mathbf{y})$ and $\boldsymbol{\Lambda}_t(\mathbf{x}, \mathbf{y})$ which are obtained by concatenating all the sampled Jacobians, residuals, and noise-precision matrices:

$$\mathbf{J}_t(\mathbf{x}) := \begin{bmatrix} \mathbf{J}_{w_t^{(1)}}(\mathbf{x}) \\ \mathbf{J}_{w_t^{(2)}}(\mathbf{x}) \\ \mathbf{J}_{w_t^{(3)}}(\mathbf{x}) \\ \vdots \\ \mathbf{J}_{w_t^{(S)}}(\mathbf{x}) \end{bmatrix}, \qquad \mathbf{r}_t(\mathbf{x}, \mathbf{y}) := \begin{bmatrix} \mathbf{r}_{w_t^{(1)}}(\mathbf{x}, \mathbf{y}) \\ \mathbf{r}_{w_t^{(2)}}(\mathbf{x}, \mathbf{y}) \\ \mathbf{r}_{w_t^{(3)}}(\mathbf{x}, \mathbf{y}) \\ \vdots \\ \mathbf{r}_{w_t^{(S)}}(\mathbf{x}, \mathbf{y}) \end{bmatrix}, \qquad (37)$$

$$\boldsymbol{\Lambda}_t(\mathbf{x}, \mathbf{y}) := \begin{bmatrix} \boldsymbol{\Lambda}_{w_t^{(1)}}(\mathbf{x}, \mathbf{y}) & 0 & 0 & \dots & 0 \\ 0 & \boldsymbol{\Lambda}_{w_t^{(2)}}(\mathbf{x}, \mathbf{y}) & 0 & \dots & 0 \\ 0 & 0 & \boldsymbol{\Lambda}_{w_t^{(3)}}(\mathbf{x}, \mathbf{y}) & \dots & 0 \\ \vdots & \vdots & & \ddots & \vdots \\ 0 & 0 & 0 & \dots & \boldsymbol{\Lambda}_{w_t^{(S)}}(\mathbf{x}, \mathbf{y}) \end{bmatrix}. \qquad (38)$$

Using this, we define a transformed output of length $KS \times 1$ as

$$\tilde{\mathbf{y}}_{i,t} := \mathbf{J}_t(\mathbf{x}_i)\boldsymbol{\mu}_t - \boldsymbol{\Lambda}_t(\mathbf{x}_i, \mathbf{y}_i)^{-1}\mathbf{r}_t(\mathbf{x}_i, \mathbf{y}_i). \qquad (39)$$

The distribution $\tilde{q}_{t+1}(\mathbf{w})$ defined in (36) can then be expressed as the following:

$$\tilde{q}_{t+1}(\mathbf{w}) \propto \mathcal{N}\left(\mathbf{w}|\mathbf{m}_t, \mathbf{V}_t\right) \prod_{i=1}^{N} \mathcal{N}\left(\tilde{\mathbf{y}}_{i,t}|\mathbf{J}_t(\mathbf{x}_i)\mathbf{w}, S(\beta_t \boldsymbol{\Lambda}_t(\mathbf{x}, \mathbf{y}))^{-1}\right). \qquad (40)$$

As before, we can show that this distribution is equal to the posterior distribution of a linear on a transformed dataset defined as $\widetilde{\mathcal{D}}_t := \{(\mathbf{x}_i, \tilde{\mathbf{y}}_{i,t})\}_{i=1}^{N}$. To model such outputs, we define a linear model for an output $\tilde{\mathbf{y}}_t \in \mathbb{R}^{KS}$ defined as follows:

$$\tilde{\mathbf{y}}_t = \mathbf{J}_t(\mathbf{x})\mathbf{w} + \boldsymbol{\epsilon}_t, \text{ with } \boldsymbol{\epsilon}_t \sim \mathcal{N}(0, S(\beta_t \boldsymbol{\Lambda}_t(\mathbf{x}, \mathbf{y}))^{-1}), \text{ and } \mathbf{w} \sim \mathcal{N}(\mathbf{m}_t, \mathbf{V}_t). \qquad (41)$$

The theorem presented in the main text is a simpler version of this theorem where $S = 1$. This completes the proof. □

### A.3 Linear Model Corresponding to OGGN

In OGGN, we evaluate the gradient and Hessian at the mean $\boldsymbol{\mu}_t$ defined to be equal to the current iterate $\mathbf{w}_t$. This corresponds to $S = 1$ in the setting described in the proof of theorem 2 (see Appendix A.2) with $\mathbf{w}_t^{(1)} := \mathbf{w}_t$. Therefore, the linear model is the same as before but with $\mathbf{J}_t(\mathbf{x}), \mathbf{r}_t(\mathbf{x}, \mathbf{y})$ and $\boldsymbol{\Lambda}_t(\mathbf{x}, \mathbf{y})$ defined at $\mathbf{w}_t$.

## B  Approximating Posterior Predictive with DNN2GP Approach

Typically, we can always predict using Monte Carlo sampling from the Gaussian approximation, however, this might be too noisy sometimes. In this section, we show how DNN2GP approach enables us to directly use the GP regression model for *approximating* the posterior predictive distribution. We elaborate on the method for Laplace approximation but this can be generalized to VI as briefly explained in subsection B.3.

Given a test input, denoted by $\mathbf{x}_*$, we first compute the feature map $\mathbf{J}_*(\mathbf{x}_*)^\top$. Using the linear model found in the DNN2GP approach, we can compute the posterior predictive distribution of the output, which we denote by $\tilde{\mathbf{y}}_*$. However, to be able to compute the predictive distribution for the true output $\mathbf{y}_*$, we need to *invert the map* from $\mathbf{y}_*$ to $\tilde{\mathbf{y}}_*$. The expressions for this map can be obtained by using the definition of the transformed output $\tilde{\mathbf{y}}_* := \mathbf{J}_*(\mathbf{x}_*)\mathbf{w}_* - \boldsymbol{\Lambda}_*(\mathbf{x}_*, \mathbf{y}_*)^{-1}\mathbf{r}_*(\mathbf{x}_*, \mathbf{y}_*)$. We demonstrate this for two common cases of squared loss and logistic loss.

### B.1  Laplace Approximation and Squared Loss

Consider the squared loss, $\ell(\mathbf{y}, \mathbf{f}_w(\mathbf{x})) = \frac{1}{2\sigma^2}\|\mathbf{y} - \mathbf{f}_w(\mathbf{x})\|^2$ with $\sigma^2$ as the noise variance. According to section 3, in this case, we have $\mathbf{r}_*(\mathbf{x}, \mathbf{y}) := \sigma^{-2}(\mathbf{f}_{w_*}(\mathbf{x}) - \mathbf{y})$ and $\boldsymbol{\Lambda}_*(\mathbf{x}, \mathbf{y}) := \sigma^{-2}\mathbf{I}_K$. Using these expressions in the definition for $\tilde{\mathbf{y}} := \mathbf{J}_*(\mathbf{x})\mathbf{w}_* - \boldsymbol{\Lambda}_*(\mathbf{x}, \mathbf{y})^{-1}\mathbf{r}_*(\mathbf{x}, \mathbf{y})$, we get the following map for the test input $\mathbf{x}_*$:

$$\tilde{\mathbf{y}}_* = \mathbf{J}_*(\mathbf{x}_*)\mathbf{w}_* - (\mathbf{f}_{w_*}(\mathbf{x}_*) - \mathbf{y}_*) \tag{42}$$
$$\implies \mathbf{y}_* = \tilde{\mathbf{y}}_* + \mathbf{f}_{w_*}(\mathbf{x}_*) - \mathbf{J}_*(\mathbf{x}_*)\mathbf{w}_* \tag{43}$$

Given a predictive distribution for $\tilde{\mathbf{y}}_*$ computed by the linear model (8) with the posterior distribution $\mathcal{N}(\mathbf{w}|\mathbf{w}_*, \widetilde{\boldsymbol{\Sigma}})$, we can therefore derive the predictive distribution for $\mathbf{y}_*$. In the example above, the predictive variance of $\tilde{\mathbf{y}}_*$ and $\mathbf{y}_*$ will be the same, while the predictive mean of $\mathbf{y}_*$ is obtained by adding $\mathbf{f}_{w_*}(\mathbf{x}_*) - \mathbf{J}_*(\mathbf{x}_*)\mathbf{w}_*$ to the mean of $\tilde{\mathbf{y}}_*$. The result is as follows

$$\mathbf{y}_*|\mathbf{x}_*, \mathcal{D} \sim \mathcal{N}\left(\mathbf{y}_*|\mathbf{f}_{w_*}(\mathbf{x}_*), \mathbf{J}_*(\mathbf{x}_*)\widetilde{\boldsymbol{\Sigma}}\mathbf{J}_*(\mathbf{x}_*)^\top + \sigma^2\mathbf{I}_K\right). \tag{44}$$

We use this technique to compute the predictive distribution in Fig. 3 (labeled as 'DNN2GP' in the plots).

### B.2  Laplace Approximation and Logistic Loss

The procedure above for *inversion of maps* generalizes to other loss functions derived using generalized linear models. We need to assume that the loss corresponds to a log probability distribution, i.e., $\ell(\mathbf{y}, \mathbf{f}_w(\mathbf{x})) := -\log p(\mathbf{y}|\mathbf{h}(\mathbf{f}_w(\mathbf{x})))$ where $\mathbf{h}(\cdot)$ is a *link function*. We now describe this for a Bernoulli distribution $y_i \in \{0, 1\}$ using the results in section 3.

Similarly to the squared-loss case, we need to write $\tilde{y}$ in terms of the true output $y$. For a Bernoulli likelihood, the link function is $\sigma(f_{w_*}(\mathbf{x})) =: p_*(\mathbf{x})$ where $\sigma$ is the sigmoid function, the residual is $r_*(\mathbf{x}, y) = p_*(\mathbf{x}) - y$, and the noise precision is $\Lambda_{w_*}(\mathbf{x}, y) = p_*(\mathbf{x})(1 - p_*(\mathbf{x})) := \lambda_*(\mathbf{x})$. We again use the definition for the transformed output and write the map for the test input $\mathbf{x}_*$:

$$\tilde{y}_* = \mathbf{J}_*(\mathbf{x}_*)\mathbf{w}_* - \lambda_*(\mathbf{x}_*)^{-1}(p_*(\mathbf{x}_*) - y_*) \tag{45}$$
$$\implies y_* = p_*(\mathbf{x}_*) + \lambda_*(\mathbf{x}_*)\tilde{y}_* - \lambda_*(\mathbf{x}_*)\mathbf{J}_*(\mathbf{x}_*)\mathbf{w}_* \tag{46}$$

Given the predictive distribution over $\tilde{y}_*$ at the test input $\mathbf{x}_*$, we can then compute the corresponding distribution over $y_*$. The predictive distribution of $\tilde{y}_*$ in the linear model (8) with the posterior distribution $\mathcal{N}(\mathbf{w}|\mathbf{w}_*, \widetilde{\boldsymbol{\Sigma}})$ is given as follows:

$$\tilde{y}_*|\mathbf{x}_*, \widetilde{\mathcal{D}} \sim \mathcal{N}\left(\tilde{y}_*|\mathbf{J}_*(\mathbf{x}_*)\mathbf{w}_*, \lambda_*(\mathbf{x}_*)^{-1} + \mathbf{J}_*(\mathbf{x}_*)\widetilde{\boldsymbol{\Sigma}}\mathbf{J}_*(\mathbf{x}_*)^\top\right). \tag{47}$$

Therefore, using the map (46), we get the following predictive distribution over $y_*$:

$$y_*|\mathbf{x}_*, \mathcal{D} \sim \mathcal{N}\left(y_*|\sigma(f_{w_*}(\mathbf{x}_*)), \lambda_*(\mathbf{x}_*) + \lambda_*(\mathbf{x}_*)^2 \mathbf{J}_*(\mathbf{x}_*)\widetilde{\boldsymbol{\Sigma}}\mathbf{J}_*(\mathbf{x}_*)^\top\right). \tag{48}$$

Similar to the linear basis function model, the two terms in the predictive variance have an interpretation (e.g., see [1] Eq. 3.59). The first term can be interpreted as the aleatoric uncertainty (label noise), while the second term takes a form that resembles the epistemic uncertainty (model noise). Such interpretation is possible due to the conversion of a DNN to a linear-bassis function model in our DNN2GP framework.

This approach can be similarly written for other Gaussian approximations. It can also be generalized to loss functions obtained using the generalized linear model. The inversion of the map is possible whenever the link function $\mathbf{h}(.)$ is invertible.

### B.3 Generalization to VI

For the VOGGN update with one MC sample, we can use the same procedure as above. The same is true for OGGN since one MC sample is replaced by the mean. For VOGGN with multiple MC samples, we get $S$ such maps. Each of those maps give us a prediction, denote it by $\tilde{\mathbf{y}}_{*,s,t}$ for sample $s$ at iteration $t$. To obtain the final prediction, we can use the average all predictions $\tilde{\mathbf{y}}_{*,s,t}$ over $s = 1, 2, 3, \ldots, S$ to get the predictive distribution for $\mathbf{y}_{*,t}$.

## C  Additional Results

In this appendix, we provide additional figures to the ones presented in Sec. 5.2.

### C.1  Further Posteriors and Kernels for MNIST and CIFAR

Fig. 7 is similar to Fig. 4a but uses the variational approximation instead of a Laplace approximation. While the posterior mean on MNIST shows very similar structure for both approximations, the kernel shows some interesting differences. There are many more negative correlations between examples from different classes in the kernel corresponding to the variational approximation. The posterior mean on CIFAR-10 has similar structure yet it appears to exhibit higher uncertainty. In Fig. 8, we show the kernel matrix on 300 data points of CIFAR-10 with the respective class labels. The kernel is computed for both the Laplace and variational approximation but shows less structure than that of the MNIST dataset.

(a) MNIST: GP kernel matrix (left) and GP posterior mean (right)    (b) CIFAR-10: GP posterior mean

Figure 7: This figure visualizes the GP kernel matrix and posterior mean for LeNet5 trained with VOGN on MNIST (left) and CIFAR-10 (right). The kernel matrix clearly shows the correlations learned by the DNN. A higher posterior mean is assigned to the correct label which reflects the accuracy obtained by the DNN.

### C.2  Uncertainties according to DNN2GP for Classification

In this section we present a toy example for the classification task in line with the regression experiment in Fig. 3. We use the reparameterization introduced in App. B.2, in particular Eq. (48).

| (a) Laplace Approximation | (b) Variational Approximation |

Figure 8: GP kernels due to Laplace and variational approximation for neural networks on CIFAR-10. The kernels show slight traces of structure but are not as significant as the ones presented on MNIST in Sec. 5.

(a) mean $\sigma(f_{w_*}(\mathbf{x}_*))$        (b) aleatoric uncertainty        (c) epistemic uncertainty

Figure 9: This figure demonstrates the decomposition of predictive variances due to the reparameterization introduced in App. B.2 on a binary toy classification task (red vs. blue half moons). We plot the quantities of Eq. (48) in figures (a)-(c): (a) is the prediction of a trained NN while the sum of (b) and (c) give us the posterior predictive uncertainties. Around the decision boundary, the label noise (b) is high and remains unchanged further from the data while the predictive uncertainty is low where supported by data and strongly grows away from it. Here, the model fits the data well in contrast to Fig. 4d where the model is unable to do so which results in high estimated label noise.

We train a neural network with single hidden layer of 10 units and tanh activation to fit the non-linear decision boundary. We have $\delta = 0.26$ and train on 100 samples for 5000 full-batch epochs. Fig. 9 shows how the reparameterization allows to decompose predictive variance into label noise due to the decision boundary, see. (b), and model uncertainty, see (c), that grows away from the data.

# D    Author Contributions Statement

Author List: Mohammad Emtiyaz Khan, Alexander Immer, Ehsan Abedi, Maciej Korzepa.

M.E.K. conceived a rough idea using the gradients and Hessians of the loss, and wrote the first version of the proofs. A.I. and E.A. made major corrections to M.E.K.'s original version and introduced version used in the final paper. They also came up with the prediction method for DNN2GP. E.A. formalized the NTK connection, and extensively studied its connection to the GP posterior. A.I. did most of the experiments and introduced the necessary reparameterization for applications. M.K. helped on the hyperparameter-tuning experiments, as well as with the visualizations. M.K. did the regression uncertainty experiment with some help from E.A. and A.I.

M.E.K. wrote the main content of the paper. E.A. wrote all the proofs, and A.I. and M.K. summarized the experiment section. All the authors proof-read the paper and revised it.

## E    Camera-Ready Version vs the Submitted Version

We made several changes taking reviewers feedback into account.

1. The writing and organization of the papers were modified to emphasize that we are able to relate the iterations of a deep-learning algorithm to GP inference.

2. To improve clarity, Fig. 1 was added as a summary of our approach. The writing was modified to follow Step A, B, and C given in Fig. 1.

3. Titles of Section 3 and 4 were changed to emphasize relationship to "solutions and iterations" of a deep-learning algorithm.

4. Theorem 1 and 2 were simplified to focus only on the posterior of linear model only. Relation to GP is discussed separately.

5. Experiment on GP regression was modified to focus on uncertainty instead of the width of the DNN.

6. Visualization of the GP predictive uncertainty and noise was added on top of predictive mean on CIFAR-10

7. A real-world experiment on Wine dataset was added, where we tune the width of the DNN.