[Reviews · NeurIPS 2019]

Reviewer 1



************ Quality ************ The technical details in the paper are sound to me. There's some space to improve for the experiments. I think the main contribution of this paper is proposing a method to transform the complicated neural network structure to a nonlinear feature mapping function, so that they can linearly separate the weight and feature mapping. Given the feature mapping, kernels/correlations and posterior distributions over output functions can be explicitly built for BNN (or DNN). Therefore, I would expect to see 1. What does this feature mapping look like? I think the authors show the kernel instead of the mapping itself. I think this kernel is NTK kernel, right? It looks very much like the correlation matrix calculated from the output of each data example. What about comparing to other kernels (as mentioned in the paper), or kernels in standard GPs? What's the influence of increasing or decreasing the number of parameters (dimensionality of the feature map)? If you just plot the correlation matrix of the output layer, do you recover similar "kernel matrices"? Without showing some comparisons, it's a bit vague to see the contribution of the connection from DNN to GP. 2. Showing the GP posterior mean only seems to be an overuse of GP since what is more interesting from GP compared with deterministic DNN is the uncertainty. 3. What's the quantitative performance (e.g. accuracy), compared with DNN, BNN, NTK, or GPs with other kernel? Better or worse, why? Figure 2 seems to be a quite obvious thing. GP with an NTK kernel but finite dimensional feature map is worse than an infinite feature map (infinitely wide NN). It's good to have it there, but more inspections for different kernels and comparisons with other BNN-GP type of works might helpful to reveal the strength or weakness of the method. I like sec. 5.3. This shows one advantage of casting DNN in GP, which helps tuning hyper parameters in a Bayesian way. ************ Clarity ************ The paper is pretty clear and well written. They did a good job establishing the connection from DNN to GP in math. ************ Originality ************ There are a few emerging works recently showing the connection between DNN and GP when the structure is infinitely wide. They get rid of the constraint and make the connection for finite hidden units, in which case the feature map in a GP has finite dimensions. They derive GPs from approximations of Hessians, which I think is pretty original. ************ Significance ************ I appreciate the attempt to understand DNN with GP posteriors and kernels. Theoretically, this helps people understand DNN. The significance would be better evaluated if the authors could add more experimental results (as mentioned in the quality part). My current evaluation is better than medium, but below high. ################### Thanks for the response from the authors which has addressed most of my questions. I think this is an interesting paper establishing the theoretical connection. What I feel missing is the experimental result to show why establishing such a connection is beneficial. Why people try to formulate BNN into GP? Tuning the hyper parameters is a good reason but the authors took a small sip of it. What about posterior inference, accuracy performance on large real datasets? The authors spent a large portion of the experiment section showing the GP kernels, which is a bit less interesting. Therefore, I would still keep my original decision.

Reviewer 2



Summary The manuscript discusses a loose relationship between GPs and nonlinear parametric models. Using a spherical Gaussian prior (i.e. l2 regularizer) on the parametric weights and a Hessian approximation in a parametric model: inference in the parametric model can be interpreted as GP inference with a degenerate GP. Originality The interpretation seems somewhat novel, but not very surprising, though. Making a local Gaussian approximation (during Laplace approximation) can certainly be interpreted as a Gaussian process in some sense. Significance The interplay between parametric models such as DNNs (and their limits) and GPs in terms of architecture and algorithms is interesting. The proposed methodology is -- however -- just a starting point and provides only little value. Quality The manuscript gives some insights but the empirical part can be strengthened. In particular, the direction of the optimization experiment in Section 5.3 is very interesting but unfortunately very short. A non-trivial demonstration or a discussion of limits, pitfalls etc. needs to be added. I'm not sure of the message that should be conveyed by Figures 3 and 4. For the paper as a whole, it does not seem that the core results are specific to DNNs but should hold for any parametric model. Also the spherical Gaussian prior seems not to be crucial. Shouldn't other smooth priors work as well? Clarity The paper is reasonably well written with some typos and room for improvement in terms of language. Details - Intro, paragraph 1: real-world problem is an important problem - Section 3, last paragraph: formulation is - Section 4, line 143: estimate Gaussian - Discussion, line 258: indicate this to certain degree - References: Capitalization (gaussian, bayesian, neural information ...) ########################### The new experiments show that the marginal likelihood can guide parameter optimization in a non-synthetic dataset and also beyond the "artificially introduced" parameter $delta$ i.e. the width of a network. However, only follow up work will have to show whether the proposed connecton between BNNs and GPs is useful beyond small datasets and and for other parameters.

Reviewer 3



Update: Thank you for the feedback. While it cleared some of my concerns, the most important one regarding the GGN approximation is not fully addressed: see my original comments, (ii) and (iii). Like the authors said, the fact that the derived models have data-dependent likelihood is helpful for understanding the behavior of approximate inference methods, but it's not so helpful in justifying their use in posterior inference. Therefore I would keep my original decision. Minor comment: it's not clear what you meant by the NTK appears for Laplace, but for VI the kernel is different''. (12) clearly looks like an NTK, evaluated at a location different from the optima. ======== This paper uses neural tangent kernel (NTK) to study BNN posterior approximations. To connect NTK to BNN posteriors, the authors consider simple Gaussian posterior approximations (Laplace and mean field VI) and further employs the GGN approximation to Hessian. This simplified setup is interesting, and should be useful for future research. However, its limitations should not be overlooked, and I wish the authors could be clearer about them. The first issue is about the justification of GGN. The authors claim (L102) that it is a good approximation because DNN fits data well. But (i) for classification problems, having a good fit does not imply the residual vanishes, e.g. when the likelihood p(y|f(x)) is parameterized by sigmoid or softmax; (ii) the neglected term is the product of residual and Hessian (\nabla^2_{ww} f); most modern network architectures use ReLU or similar activations, which leads to very large (in theory, infinity for ReLU) Hessian, so a small residual does not necessarily lead to small approximation errors; (iii) in the analysis of VI (Eq (11)), it is not even clear if models in the high probability region in the variational distribution all have a good fit. Therefore, the authors should be clear at the beginning that they are considering the GGN approximation, and either provide empirical evidence that GGN is a good approximation to the original Hessian (especially on classification problems and for ReLU-like activations), or show that posterior approximations with GGN still have good performance. Secondly, all derived GP models have data-dependent likelihood models (\epsilon \sim N(0, \Lambda^{-1})). The only exception is in least-square regression problems, in which \Lambda is independent of data and model. This is not similar to standard Bayesian modeling practice, and makes the results a lot less appealing when the original BNN problem has non-Gaussian likelihood. While it is not deal-breaking, I believe it is necessary that the authors acknowledge this limitation, or provide justifications that the derived GP models are still suitable in statistical modeling. Apart from the aforementioned issues, the paper is well written and the derivations appear correct.

[Author Response · NeurIPS 2019]

We thank the reviewers for their feedback. A big concern among all reviewers is about
experimental results. We emphasize that our main contribution is to derive theoretical
connections, but, as per your suggestions, we will add the following new experiments:

1. Hyperparameter tuning for discrete parameters (see top figure on the right on
choosing the NN width) and on a real dataset (see bottom figure on the right for setting
prior-precision on the "UCI Wine" dataset). We also have results on comparing different
architectures (LeNet, AlexNet, ResNet) on CIFAR, which we will add in the paper.
2. Feature maps visualization on real data in the appendix since this takes a lot of space.
3. Comparisons with other kernels and with other BNN-GP method on a small example.

There are also a few concerns by R2 regarding originality and significance of our work.
We would like to emphasize that this is the first result connecting training procedures
and stationarity conditions of BNNs to GP inference. In particular, no other existing
work has been able to express iterations of a VI procedure as GPs (Theorem 3). We
agree that this paper takes the first step, but it is an important step.

R3 has some concerns about the GGN approximation, but these have mostly been
resolved by other recent works. We have provided an explanation in the response to R3.

*R1: what does the feature mapping look like?* - We show an example for the toy data in Fig (1b) in the paper. For real
datasets, these are too big to visualize which is why we only show kernels. We will add a visualization in the appendix.
*R1: Your NTK kernel looks very much like the correlation matrix of the output of each data example. What about*
*comparing to other kernels or kernels in standard GPs?* - The NTK kernel is built using Jacobians, i.e., by using the
first-order information, which is fundamentally different from other kernels used in GP. We will add visualizations
of various kernels in the appendix to show a comparison. Our kernel can be seen as an approximation to the output
correlation matrix.
*R1: What's the influence of increasing or decreasing the number of parameters?* - Increasing the number of parameters
can capture complicated information, but then the marginal likelihood penalizes for the increase in number of parameters.
This trade-off is clear when we plot it with respect to the network width (see the figure on the right).
*R1: when using GP, uncertainty should be shown.* - We have these results and will add them in the paper. The GP
uncertainty is in line with that of Bayesian NN uncertainty obtained by sampling from the posterior approximation.
*R1: how about quantitative performance compared to other models and BNN-GP relations? This may reveal the*
*strength or weakness of the method.* - the performance of resulting GP is equal to that of a BNN, so this comparison is
not necessary. Other BNN-GP methods are computationally demanding since they require computation and inversion of
the kernel, which is why we are restricted to a toy problem (Fig. 2). We will try to add a more realistic example.

*R2: Laplace approximation can be certainly interpreted as GP in some way?* - It might appear that it is easy to derive
this connection explicitly, but until now there are no such results. Our derivation also extends to VI where *every iteration*
can be expressed a GP. This result is nontrivial and first of its kind.
*R2: Also the spherical Gaussian prior seems not to be crucial. Shouldn't other smooth priors work as well?* - This is
correct and the method works even for nonsmooth priors such as Laplace. We will emphasize this in the paper.
*R2: Provide more insights into algorithmic challenges such as runtime, numerics etc.* We will add this in the text.

*R3: This paper uses neural tangent kernel (NTK) to study BNN posterior approximations.* - It appears that there is a
misunderstanding here. The goal is to show that by using approximate posteriors we recover a GP. The NTK appears
for Laplace, but for VI the kernel is different.
*R3: For classification problems, the residuals do not vanish.* - This is not entirely correct. Residuals are gradients of the
loss and they tend to zero as the network classification for a data example becomes better and better. See *New insights*
*and perspectives on the natural gradient method* (Martens, 2014).
*R3: Provide empirical evidence that posterior approximation with GGN have good performance.* - Recent works have
clearly shown that GGN based VI algorithms work well; see *Practical Variational Inference for Neural Networks*
(Graves, 2011), *Noisy Natural Gradient as Variational Inference* (Zhang, 2017), *Fast and scalable Bayesian deep*
*learning by weight-perturbation in Adam* (Khan, 2018) . We will add a discussion on the accuracy of GGN referring to
these papers.
*R3: all derived GP models have data-dependent likelihood models and authors should acknowledge this limitation* - It is
incorrect to say that this is a limitation of the method. Such data-dependent likelihood "approximations" are in fact very
common and arise in methods such as: iterative weight least squares, expectation propagation, and even in well known
variational bounds such as Jordan and Jaakkola's bound (see Bishop's book). For example, when approximating a binary
likelihood, such data-dependent approximations are essential where variance is adjusted to get better approximations.
This is not a limitation but an advantage that helps us to figure out important data examples, e.g. boundary points in a
classification problem.

[Meta-Review · NeurIPS 2019]

This paper demonstrates theoretically that multiple forms of approximate Bayesian inference (Laplace approximation and variational inference) for deep neural networks are equivalent to Gaussian processes. The authors formalize this connection and write out the GP covariance function corresponding to these networks, which surprisingly turns out to be the neural tangent kernel. The authors also establish a connection to the training procedure of the neural network and GPs, which is a novel contribution. There is a growing literature on the connection between neural networks and Gaussian processes, with a variety of papers establishing the connection in the infinite limit of hidden units. This paper adds nicely to that literature, developing a connection to approximate Bayesian inference. The reviewers found the paper insightful and sensible. However, their main concern was with respect to empirical evaluation as they found the experiments underwhelming. However, the experiment involving optimizing the hyperparameters of a DNN via the marginal likelihood of its corresponding GP is a neat and novel concept. The recommendation is for acceptance as it is believed that this work provides some interesting insights and connections and opens up a variety of avenues of future work.